# Finite-Time Disturbance Observer-Based Adaptive Course Control for Surface Ships

**DOI:** 10.3390/s24154843

**Published:** 2024-07-25

**Authors:** Ming Xu, Chenglong Gong

**Affiliations:** 1School of Automation, Wuhan University of Technology, Wuhan 430070, China; whutxuming@whut.edu.cn; 2School of Information Engineering, Fujian Polytechnic of Water Conservancy and Electric Power, Yong’an 366000, China

**Keywords:** course tracking, input saturation, disturbance observer, hyperbolic tangent, LaSalle’s Invariance Principle

## Abstract

In this paper, a finite-time disturbance observer-based adaptive control strategy is proposed for the ship course control system subject to input saturation and external disturbances. Based on the Gaussian error function, a smooth saturation model is designed to avoid the input saturation of the system and reduce steering engine vibrations, and an auxiliary dynamic system is introduced to compensate for the effect of the rudder angle input inconsistency on the system. By constructing an auxiliary dynamic, a finite-time disturbance observer is designed to approximate the external disturbance of the system; an adaptive updating law is also constructed to estimate the upper bound of the derivative of the external disturbance. Combining the finite-time disturbance observer with the auxiliary dynamic system, a novel adaptive ship course control law is proposed by using the hyperbolic tangent function. Moreover, according to LaSalle’s Invariance Principle, a system stability analysis method with loose stability conditions and easy realizations is designed, while the stability of the closed-loop system and the ultimately uniformly boundedness of all its signals are proven. Finally, the course control simulation analysis of a surface ship is carried out. The results show that the proposed control law has a strong resistance to external disturbances and a strong non-fragility to system parameter perturbations, which ensure that the course control system has great control performance.

## 1. Introduction

Course control [1] is a fundamental aspect of ship motion control, aiming at maintaining the desired heading or tracking new headings when a vessel is subjected to external disturbances. Researching robust and reliable ship course control systems holds significant practical importance and profound implications for the development of deep-sea exploration and the marine economy. The ship course control system continuously monitors the vessel’s heading and external disturbances using sensors and a disturbance observer, as shown in Figure 1, where $\psi_{d}$ is the desired constant heading, ψ represents the heading angle, $\psi_{e}=\psi -\psi_{d}$; $\delta_{c}$ is the command control signal of the rudder angle, δ stands for the actual rudder angle, and d represents the external disturbances to the system.

With the advancement of control technology, various advanced nonlinear control techniques have been developed and effectively implemented for controlling ship heading and motion [2,3,4,5,6]. However, the research on ship heading control still faces challenges such as model parameter perturbations, external disturbances, and input saturation.

During ship navigation, it is inevitable that the vessel will be influenced by disturbances caused by wind, waves, and currents. Chen et al. [4] introduced a novel adaptive controller incorporating nonlinear modulators to enhance adaptability and rudder efficiency, thereby achieving satisfactory course-tracking performance for ships. Quang Dich et al. [7] proposed a novel disturbance observer designed to estimate compromised signals on public channels and the uncertain values within the secure communication system. Fu et al. [8] introduced a model predictive control approach utilizing an enhanced nonlinear disturbance observer to achieve satisfactory motion control performance for USVs. Lei and Guo [3] presented a robust ship heading controller that compensated for external uncertainties and the yaw angle. Additionally, neural networks [9,10] and fuzzy control [11,12] have also been employed to handle external disturbances in heading control systems. However, the online updating of weight matrices in these methods increases the computational burden of the system. It should be noted that all the disturbance handling methods mentioned above do not guarantee finite-time convergence of disturbance estimation errors.

Input saturation constraints critically impact the performance of nonlinear control systems and can potentially cause instability. Tang et al. [13] investigated the adaptive optimized leader–follower consensus control issue for discrete-time multi-agent systems characterized by asymmetric input saturation constraints and hybrid faults, employing an optimized Backstepping technique. Moreover, Gaussian functions were employed to handle input saturation in the heading control system. Combined with the Backstepping technique, Bai et al. [14] designed an optimized heading control method. However, the Backstepping technique may lead to the issue of “dimensional explosion” in the system. Hu et al. [15] developed a dynamic saturation filter to manage input saturation in the system and compensated for the adverse effects of input deviation on ship heading control.

Inspired by the aforementioned literature, this study proposes a novel approach for ship heading control under external disturbances and input saturation conditions. A ship adaptive heading control law based on a finite-time disturbance observer is designed, which not only achieves the accurate tracking of ship heading under constraints but also ensures the ultimate boundedness of all signals in the system. The key innovations of this paper are as follows:
(1)Through the construction of auxiliary dynamics, a finite-time disturbance observer is developed to appraise time-varying external disturbances, ensuring that the disturbance estimation error converges in finite time. Moreover, by devising a self-tuning update law to estimate the maximum rate of the external disturbances’ derivative, the conservatism in the design of the disturbance observer is reduced.(2)By the combination of the finite-time disturbance observer and auxiliary dynamics, an adaptive heading control law based on the hyperbolic tangent function is developed to achieve accurate tracking of the ship’s heading. Furthermore, leveraging LaSalle’s Invariance Principle, a simple and effective method for system stability analysis is designed. Unlike traditional Lyapunov function-based proof methods, this stability condition is very lenient and simple to meet. Even if the scalar function used to demonstrate system stability is non-positive definite with a semi-negative definite derivative, it still ensures the long-term stability of the system.

The notations used in this study include the following: $\left|\cdot \right|$ represents the absolute value of a constant or variable; $\lambda_{\min}(\cdot )$ and $\lambda_{\max}(\cdot )$ represent the minimum and maximum eigenvalues of a matrix; ln(⋅) represents the logarithmic function; $\left\|\cdot \right\|$ denotes the second norm of a matrix; sign(⋅) represents the sign function; tanh(⋅) corresponds to the hyperbolic tangent; cosh(⋅) represents the hyperbolic cosine; min(⋅) denotes the selection of the minimum value within (⋅); and $erf(x)=2\int_{0}^{x}e^{-\eta^{2}}d\eta /\sqrt{\pi }$ signifies the Gaussian error function, as shown in Appendix A.

## 2. Prior Knowledge and Problem Statement

### 2.1. Prior Knowledge

**Lemma** **1.**
*LaSalle’s Invariance Principle [16] takes into account a self-governing system described by*



(1)
$$\;\dot{x}=f(x)$$


Given that $f:D\to R^{n}$ is a local Lipschitz mapping from the domain D to $R^{n}$, and there exists a continuously differentiable scalar function V:D→R within the set Ω that satisfies condition $\dot{V}(x)\leq 0$. Furthermore, Ω⊂D represents a positive invariant compact set for the equation $\dot{x}=f(x)$. Assuming that E is a set comprising all points within Ω that satisfies condition $\dot{V}(x)=0$, and let M be the largest invariant set within E, then, when t→∞, every solution starting from the set Ω tends towards M.

**Remark** **1.***LaSalle’s Invariance Principle extends the Lyapunov direct method, which also involves the use of a scalar function* V(x) *to prove system stability. Both methods require the scalar function* V(x) *to decay, with the condition* $\dot{V}(x)\leq 0$*. However, there is a difference between the two approaches. Within the Lyapunov direct method, the scalar function* V(x) *needs to be positively definite, whereas, in LaSalle’s Invariance Principle, it can be non-positively definite. Another significant benefit of LaSalle’s Invariance Principle is its applicability even in cases where the derivative of the scalar function shows semi-negative definiteness, as the system continues to guarantee asymptotic stability. Therefore, LaSalle’s Invariance Principle allows for relatively relaxed and easily satisfied stability conditions for the system.*

**Lemma** **2.***If a dynamical system with a hyperbolic tangent function has the following form [17]*(2)$$\left\{\begin{array}{l} {\dot{\gamma }}_{1}=\gamma_{2}\\ {\dot{\gamma }}_{2}=-\alpha \tanh(k\gamma_{1}+l\gamma_{2})-\beta \tanh(l\gamma_{2}) \end{array}\right.$$*where* α,β,k,l>0*, then the system is globally asymptotically stable*.

**Proof:** Considering that $\cosh(x)=\left(e^{x}+e^{-x}\right)/2\geq 1$ and $\ln\left(\cosh(x)\right)\geq 0$, $\ln\left(\cosh(x)\right)=0$ if and only x=0. To prove as t→∞, $\gamma_{1}\to 0$ and $\gamma_{2}\to 0$, we construct a scalar function V as
(3)$$V=\alpha \ln\left(\cosh(k\gamma_{1}+l\gamma_{2})\right)+\beta \ln\left(\cosh(l\gamma_{2})\right)+\frac{1}{2}k\gamma_{2}^{2}$$Taking the derivative of V, we obtain that
(4)$$\dot{V}=\alpha \frac{\sinh(k\gamma_{1}+l\gamma_{2})}{\cosh(k\gamma_{1}+l\gamma_{2})}(k{\dot{\gamma }}_{1}+l{\dot{\gamma }}_{2})+\beta \frac{\sinh(l\gamma_{2})}{\cosh(l\gamma_{2})}l{\dot{\gamma }}_{2}+k\gamma_{2}{\dot{\gamma }}_{2}$$Define $t_{1}=\tanh(k\gamma_{1}+l\gamma_{2})$ and $t_{2}=\tanh(l\gamma_{2})$; thus, (2) can be rewritten as ${\dot{\gamma }}_{2}=-\alpha t_{1}-\beta t_{2}$, and consequently, (4) can be expressed as
(5)$$\dot{V}=-l{\left(\alpha t_{1}+\beta t_{2}\right)}^{2}-k\beta \gamma_{2}t_{2}$$Since $x\tanh(x)=x(e^{x}-e^{-x})/\left(e^{x}+e^{-x}\right)\geq 0$, it follows that $\gamma_{2}t_{2}=\gamma_{2}\tanh(l\gamma_{2})\geq 0$ and $\dot{V}\leq 0$. $\dot{V}=0$ if and only $\gamma_{1}=\gamma_{2}=0$. In accordance with LaSalle’s Invariance Principle, system (2) is asymptotically stable, that is, as t→∞, $\gamma_{1}\to 0$ and $\gamma_{2}\to 0$. The rate at which the system converges hinges on α,β,k,l. Lemma 2 is thus proven. □

### 2.2. Problem Statement

The Mathematical Model of Responsive Ship Nonlinear Course Control [2,15] can be expressed as
(6)$$T\ddot{\psi }+\dot{\psi }+a{\dot{\psi }}^{3}=K(\delta +\delta_{w})$$
where ψ represents the heading angle; δ stands for the actual rudder angle; $\delta_{w}$ represents the equivalent rudder angle caused by marine environmental disturbances; and T,a,K are model parameters. The state variables are selected as $x_{1}=\psi$, $x_{2}={\dot{x}}_{1}=r=\dot{\psi }$, where r is the yaw rate; then
(7)$$\left\{\begin{array}{l} {\dot{x}}_{1}=x_{2}\\ {\dot{x}}_{2}=f(x)+u+d \end{array}\right.$$
where $f(x)=-\left(\dot{\psi }+a{\dot{\psi }}^{3}\right)/T$; u=Bδ; B=K/T; and $d=B\delta_{w}$ represent the external disturbances to the system. The input saturation caused by the physical characteristics of the steering gear is represented as:(8)$$\delta =\left\{\begin{array}{c} \delta_{\max}sign(\delta_{c}),\left|\delta_{c}\right|>\delta_{\max}\\ \delta_{c},\left|\delta_{c}\right|\leq \delta_{\max} \end{array}\right.$$
where $\delta_{c}$ is the command control signal of the rudder angle and $\delta_{\max}$ is the saturation amplitude.

Because of the non-smooth characteristics of (8), it is not suitable for direct application in the design of the course control law. To reduce chattering and achieve smooth steering, this paper designs a smooth saturation model based on Gaussian basis functions.
(9)$$\delta =\delta_{M}\times erf\left(\left(\delta_{c}\sqrt{\pi }\right)/2\delta_{M}\right)$$
where $\delta_{M}=\delta_{\max}\cdot sign(\delta_{c})$.

Taking the function $\delta_{c}=t$ as the example and choosing $\delta_{\max}=6$, simulations are carried out using both model (8) and (9), respectively. The simulation results displayed in Figure 2 suggest that model (9) can smoothly approximate the saturation model (8), which provides a guarantee for smooth steering.

Define $\Delta \delta =\delta -\delta_{c}$, which represents the input deviation between the commanded rudder angle and the actual rudder angle. Then, (7) can be expressed as
(10)$$\left\{\begin{array}{l} {\dot{x}}_{1}=x_{2}\\ {\dot{x}}_{2}=f(x)+u_{c}+\Delta u+d \end{array}\right.$$
where $u_{c}=B\delta_{c};\Delta u=B\Delta \delta$ and $u=u_{c}+\Delta u$.

**Remark** **2.**
*It is important to note that input deviation is inevitable when the steering gear experiences input saturation or when the control system uses the smooth model (9) to process the control input for the rudder angle. This deviation can adversely affect the control performance of the system. Therefore, it is necessary to compensate for the input deviation to enhance the system’s control performance.*


**Assumption** **1.***The time-varying external disturbance acting on the ship’s course control system caused by the marine environment is the superposition of finitely many sinusoidal components, it is bounded [15], which means*(11)$$\left\|d(t)\right\|=\left\|\sum_{i=1}^{q}a_{i}\sin(\omega_{i}t+\sigma_{i})\right\|\leq \phi$$*where the unknown amplitudes* $a_{i}\in R$*, the unknown frequencies* $\omega_{i}\in R$*, and the unknown phases* $\sigma_{i}\in R(i=1,2\ldots \ldots q)$*,* 0<ϕ<∞*, where* q *is the number of finitely many sinusoidal components.*

**Remark** **3.**
*In general, the ocean has limited energy, and the energy possessed by the external disturbances acting on the ship is also limited. Therefore, the external disturbances affecting the ship’s course control system are bounded. Thus, Assumption 1 holds.*


The aim of this paper is to develop a self-adjusting control strategy for the ship’s course control system, which is subject to external disturbances and input saturation constraints, to achieve accurate control of the heading and to ensure the ultimate boundedness of every signal within the closed-loop system.

## 3. Adaptive Course Control Law Design

In this section, we first design a finite-time disturbance observer to assess the unknown outside disturbances and prove the finite-time convergence of the disturbance estimation error. Next, we build an auxiliary dynamic system to mitigate the negative impacts of rudder angle input variations affecting the system. Then, a self-adjusting control strategy is formulated using the hyperbolic tangent function, and it is proven that the closed-loop system under this control strategy exhibits stability and that every inside signal remains ultimately consistently bounded by using LaSalle’s Invariance Principle. The adaptive course control system for the ship is displayed in Figure 3.

### 3.1. Finite-Time Disturbance Observer

Design the integral sliding surface as
(12)$$s=x_{2}-\int_{0}^{t}z\;dt$$
where z is the auxiliary dynamic, which satisfies
(13)$$z=f(x)+u+\hat{d}+\lambda_{0}s+v$$
where $\hat{d}$ is the estimate of d, and $v=\lambda_{1}sign(s)$, $\lambda_{0}$ and $\lambda_{1}$ are constants to be designed.

Based on (7), (12), and (13), we can obtain
(14)$$\dot{s}=\tilde{d}-\lambda_{0}s-v$$
where $\tilde{d}=d-\hat{d}$.

Define σ as the maximum rate of the derivative of the external disturbance of $\dot{d}$. To reduce the conservatism in the design of the disturbance observer, construct an adaptive update law for the estimate σ, which is expressed as
(15)$$\dot{\hat{\sigma }}=-\delta_{0}\hat{\sigma }+2\left|v\right|$$
where $\hat{\sigma }$ is the estimate of σ; $\delta_{0}$ is a positive constant.

Design the finite-time disturbance observer as follows
(16)$$\left\{\begin{array}{l} \hat{d}=\xi +\lambda_{2}x_{2}\\ \dot{\xi }=\lambda_{2}(-f(x)-u-\hat{d})+(\hat{\sigma }+\lambda_{3})sign(v) \end{array}\right.$$
where ξ is an intermediate control variable; $\lambda_{2}$ and $\lambda_{3}$ are constants to be designed.

Based on (15) and (16), we can obtain
(17)$$\begin{array}{ll} \dot{\tilde{d}} & =(d-\hat{d}{)}^{\prime }=\dot{d}-\dot{\hat{d}}=\dot{d}-(\xi +\lambda_{2}x_{2}{)}^{\prime }=\dot{d}-(\dot{\xi }+\lambda_{2}{\dot{x}}_{2})\\ & =\dot{d}-[\lambda_{2}(-f(x)-u-\hat{d})+(\hat{\sigma }+\lambda_{3})sign(v)+\lambda_{2}{\dot{x}}_{2}]\\ & =\dot{d}-[\lambda_{2}({\dot{x}}_{2}-f(x)-u-\hat{d})+(\hat{\sigma }+\lambda_{3})sign(v)]\\ & =\dot{d}-[\lambda_{2}(\dot{s}+z-f(x)-u-\hat{d})+(\hat{\sigma }+\lambda_{3})sign(v)]\\ & =\dot{d}-[\lambda_{2}(\dot{s}+\lambda_{0}s+v)+(\hat{\sigma }+\lambda_{3})sign(v)]\\ & =\dot{d}-\lambda_{2}\tilde{d}-(\hat{\sigma }+\lambda_{3})sign(v) \end{array}$$

**Theorem** **1.**
*Consider the auxiliary dynamics (13) and the finite-time disturbance observer (16), if the selection of the parameters to be designed satisfies*

(18)
$$\lambda_{0}>0,\lambda_{1}\geq \left|\hat{d}\right|+\gamma +\left|d\right|,\lambda_{2}>0,\lambda_{3}>0$$

*where γ is a positive invariant, then the disturbance estimation error $\tilde{d}$ can converge within a finite period to a bounded closed set that includes the equilibrium point.*


**Proof:** Choose the Lyapunov function as
(19)$$V_{1}=s^{2}$$Calculate the time derivative of (19), and according to (14), Young’s inequality, and $v=\lambda_{1}sign(s)$, we can obtain
(20)$$\begin{array}{ll} {\dot{V}}_{1} & =2s\dot{s}\\ & \leq -2\lambda_{0}{\left|s\right|}^{2}-2(\lambda_{1}-\left|\tilde{d}\right|)\left|s\right|\\ & \leq -2(\lambda_{1}-\left|\tilde{d}\right|)\left|s\right| \end{array}$$Based on $\left|\tilde{d}\right|\leq \left|d\right|+\left|\hat{d}\right|$ and (18), (20) can be rewritten as
(21)$${\dot{V}}_{1}\leq -2\gamma V_{1}^{\frac{1}{2}}$$Based on the finite-time stability theory of systems in [18] and (21), it can be derived that the sliding mode variable s converges to zero within a finite time $t_{1}\leq \frac{V_{1}^{\frac{1}{2}}(0)}{\gamma }$, where $V_{1}(0)$ represents the initial value of $V_{1}$. Thereafter, $\dot{s}=0$ holds true constantly. Then, according to the equivalent output principle in [19] and (14), $\tilde{d}$ is equivalent to $\lambda_{1}sign(s)$ after $t_{1}$.Continue to prove that the disturbance observer is stable and the estimation error is bounded. Build the Lyapunov function as
(22)$$V_{2}={\tilde{d}}^{2}+\frac{1}{2}{\tilde{\sigma }}^{2}$$
where $\tilde{\sigma }=\sigma -\hat{\sigma }$.According to (17), calculate the time derivative of (22) as
(23)$$\begin{array}{ll} V_{2} & =2\tilde{d}\dot{\tilde{d}}-\tilde{\sigma }\dot{\hat{\sigma }}\\ & =-2\lambda_{2}{\tilde{d}}^{2}+2\tilde{d}\dot{d}-2\lambda_{3}\tilde{d}sign(v)-2\hat{\sigma }\tilde{d}sign(v)-\tilde{\sigma }\dot{\hat{\sigma }} \end{array}$$Since σ is the maximum rate of the derivative of the external disturbance $\dot{d}$, then $\sigma \geq \dot{d}$, and combining the equivalent transformation ${\left(\tilde{d}\right)}_{eq}=v=\lambda_{1}sign(s)$ after a finite time $t_{1}$, by substituting (15) into (23), we can obtain
(24)$$\begin{array}{ll} {\dot{V}}_{2} & \leq -2\lambda_{2}{\tilde{d}}^{2}+2\sigma \left|\tilde{d}\right|-2\lambda_{3}\tilde{d}\mathrm{sign}(\tilde{d})-2\hat{\sigma }\tilde{d}\mathrm{sign}(\tilde{d})+\delta_{0}\tilde{\sigma }\hat{\sigma }-2\tilde{\sigma }\left|v\right|\\ & \leq -2\lambda_{2}{\tilde{d}}^{2}-2\lambda_{3}\left|\tilde{d}\right|+2\tilde{\sigma }\left|d\right|+\delta_{0}\tilde{\sigma }\hat{\sigma }-2\tilde{\sigma }\left|v\right|\\ & \leq -2\lambda_{3}\left|\tilde{d}\right|+\delta_{0}\tilde{\sigma }\hat{\sigma } \end{array}$$According to Young’s inequality, we know
(25)$$\delta_{0}\tilde{\sigma }\hat{\sigma }=\delta_{0}\tilde{\sigma }(\sigma -\tilde{\sigma })\leq \delta_{0}\left(\frac{\sigma^{2}}{2}-\frac{{\tilde{\sigma }}^{2}}{2}\right)$$By substituting (25) into (24), we can obtain
(26)$$\begin{array}{ll} {\dot{V}}_{2} & \leq -2\lambda_{3}\left|\tilde{d}\right|+(\frac{\delta_{0}}{2}\sigma^{2}-\frac{\delta_{0}}{2}{\tilde{\sigma }}^{2})=-2\lambda_{3}\left|\tilde{d}\right|-\frac{\delta_{0}}{2}{\tilde{\sigma }}^{2}+\frac{\delta_{0}}{2}\sigma^{2}\\ & =-2\lambda_{3}\left|\tilde{d}\right|-\frac{\delta_{0}}{2}{\left({\tilde{\sigma }}^{2}\right)}^{\frac{1}{2}}+\frac{\delta_{0}}{2}{\left({\tilde{\sigma }}^{2}\right)}^{\frac{1}{2}}-\frac{\delta_{0}}{2}{\tilde{\sigma }}^{2}+\frac{\delta_{0}}{2}\sigma^{2} \end{array}$$Since $\frac{\delta_{0}}{2}{\left({\tilde{\sigma }}^{2}\right)}^{\frac{1}{2}}-\frac{\delta_{0}}{2}{\tilde{\sigma }}^{2}+\frac{\delta_{0}}{2}\sigma^{2}$ has an upper bound $\vartheta_{0}=\frac{\delta_{0}+4\delta_{0}\sigma^{2}}{8}$, then (26) can be rewritten as
(27)$$\begin{array}{ll} {\dot{V}}_{2} & \leq -\min\left\{2\lambda_{3},\frac{\delta_{0}}{\sqrt{2}}\right\}\left(\left|\tilde{d}\right|+\frac{1}{\sqrt{2}}{\left({\tilde{\sigma }}^{2}\right)}^{\frac{1}{2}}\right)+\vartheta_{0}\\ & \leq -\min\left\{2\lambda_{3},\frac{\delta_{0}}{\sqrt{2}}\right\}V_{2}^{\frac{1}{2}}+\vartheta_{0} \end{array}$$According to [20], the estimation error could converge to a bounded closed set $\Omega_{0}=\left\{\tilde{d}\in R^{n}\left|\left|\tilde{d}\right|\leq \frac{\vartheta_{0}}{(1-\rho )\min\left\{2\lambda_{3},\frac{\delta_{0}}{\sqrt{2}}\right\}}\right.\right\}$ within a finite time $t_{2}=t_{1}+\frac{2V_{2}^{\frac{1}{2}}(t_{1})}{\rho \min\left\{2\lambda_{3},\frac{\delta_{0}}{\sqrt{2}}\right\}}$, where 0<ρ<1. Theorem 1 is thus proven. □

**Remark** **4.**
*The disturbance observer designed in this paper can effectively approximate external disturbances by reducing the energy consumption in ship course control and ensuring finite-time convergence of disturbance estimation errors. Moreover, the conservatism in the prior knowledge requirements for external disturbances in the course control law design is diminished by the adaptive update law (15), which estimates the unknown upper bound of the disturbance derivative.*


**Remark** **5.***The discontinuity of the sign function may cause high-frequency chattering in the system. By using continuous functions* $\frac{s}{\left|s\right|+L_{1}}$ 
*and* $\frac{v}{\left|v\right|+L_{2}}$ *to replace the sign functions in (13) and (16), the chattering of the system can be effectively reduced, thereby improving the system’s control performance.*

### 3.2. Adaptive Course Control System

#### 3.2.1. Auxiliary Dynamic System

To offset the negative impacts of input deviation Δu on the system, this paper introduces an auxiliary dynamic system, which can be expressed as
(28)$$\dot{\theta }=-k_{3}\theta +\Delta u$$
where $k_{3}>0$ is the parameter to be designed and θ is the state vector of the auxiliary dynamic system.

In order to prove that the auxiliary dynamic system is stable, a Lyapunov function is chosen as:(29)$$V_{3}=\frac{1}{2}\theta^{2}$$

According to (28) together with Young’s inequality, the time derivative of (29) is calculated as
(30)$$\begin{array}{ll} {\dot{V}}_{3} & =\theta (-k_{3}\theta +\Delta u)\\ & \leq -k_{3}\theta^{2}+\frac{1}{2}\theta^{2}+\frac{1}{2}\Delta u^{2}\\ & \leq -2pV_{3}+\kappa \end{array}$$
where $p=k_{3}-\frac{1}{2}$; $\kappa =\frac{1}{2}\Delta u^{2}$.

In order to ensure that the auxiliary dynamic system is stable, the design parameter $k_{3}$ must satisfy the following condition
(31)$$k_{3}>\frac{1}{2}$$

Solving (30), we can obtain
(32)$$\left|\theta \right|\leq \sqrt{\kappa /2p+\left[V_{3}(0)-\kappa /2p\right]e^{-2pt}}$$
where $V_{3}(0)$ is the initial value of $V_{3}$. Therefore, θ is ultimately uniformly bounded.

To facilitate the subsequent system stability analysis, the following assumption is introduced. The rationality of the assumption will be explained later.

**Assumption** **2.***The first derivative of the state vector* θ *is bounded, that is*(33)$$\left|\dot{\theta }\right|\leq \varepsilon_{1}$$*where* $\varepsilon_{1}>0$*.*

#### 3.2.2. Adaptive Course Control Law Based on Hyperbolic Tangent

Define $\psi_{e}=\psi -\psi_{d}$, where $\psi_{d}$ is the desired constant heading, so ${\dot{\psi }}_{e}=\dot{\psi }$; and define $e_{1}=\psi_{e}$ and $e_{2}={\dot{\psi }}_{e}$. According to (10), we can obtain
(34)$$\left\{\begin{array}{l} {\dot{e}}_{1}=e_{2}\\ {\dot{e}}_{2}=f(x)+u_{c}+\Delta u+d \end{array}\right.$$

According to Lemma 2, the adaptive course control law based on the hyperbolic tangent function is designed as
(35)$$u_{c}=-\alpha \tanh(ke_{1}+le_{2})-\beta \tanh(le_{2})-\hat{d}-f(x)-k_{3}\theta$$
where α,β,k,l are all positive constants.

By substituting (35) into (34), and according to (28), we can obtain
(36)$${\dot{e}}_{2}=-\alpha \tanh(ke_{1}+le_{2})-\beta \tanh(le_{2})+\tilde{d}+\dot{\theta }$$

**Theorem** **2.**
*Under the conditions of Assumption 1 and Assumption 2, for the ship heading mathematical model (6), the course control law (35) designed with the finite-time disturbance observer (16) and the auxiliary dynamic system (28) can ensure the conclusions below:*
(1)
*The closed-loop system maintains stability;*
(2)
*Every signal within the closed-loop system remains ultimately consistently bounded;*
(3)
*The heading tracking error can reach a bounded closed set within a finite time.*



**Proof:** Choose a scalar function as
(37)$$V_{4}=\alpha \ln\left(\cosh(ke_{1}+le_{2})\right)+\beta \ln\left(\cosh(le_{2})\right)+\frac{1}{2}ke_{2}^{2}$$Taking the derivative of (37), we can obtain
(38)$$\begin{array}{ll} {\dot{V}}_{4} & =\alpha \frac{\sinh(ke_{1}+le_{2})}{\cosh(ke_{1}+le_{2})}(k{\dot{e}}_{1}+l{\dot{e}}_{2})+\beta \frac{\sinh(le_{2})}{\cosh(le_{2})}l{\dot{e}}_{2}+ke_{2}{\dot{e}}_{2}\\ & =\alpha \tanh(ke_{1}+le_{2})(k{\dot{e}}_{1}+l{\dot{e}}_{2})+\beta \tanh(le_{2})l{\dot{e}}_{2}+ke_{2}{\dot{e}}_{2} \end{array}$$Define $t_{1}=\tanh(ke_{1}+le_{2})$, $t_{2}=\tanh(le_{2})-\frac{1}{\beta }\tilde{d}+\frac{1}{\beta }\dot{\theta }$, and then ${\dot{e}}_{2}=-\alpha t_{1}-\beta t_{2}$; thus,
(39)$$\begin{array}{ll} \left|{\dot{e}}_{2}\right| & =\left|-\alpha t_{1}-\beta t_{2}\right|\\ & \leq \alpha \left|\tanh(ke_{1}+le_{2})\right|+\beta \left|\tanh(le_{2})-\frac{1}{\beta }\tilde{d}+\frac{1}{\beta }\dot{\theta }\right|\\ & \leq \alpha +\beta +\left|\tilde{d}\right|+\left|\dot{\theta }\right| \end{array}$$Combining (33) and (39), (38) can be further expressed as:(40)$$\begin{array}{ll} {\dot{V}}_{4} & \leq \alpha \left(ke_{2}+l(-\alpha t_{1}-\beta t_{2})\right)t_{1}+\beta l(-\alpha t_{1}-\beta t_{2})\left(t_{2}+\frac{\tilde{d}}{\beta }-\frac{\dot{\theta }}{\beta }\right)+ke_{2}(-\alpha t_{1}-\beta t_{2})\\ & =\alpha ke_{2}t_{1}-l\alpha^{2}t_{1}^{2}-2l\alpha \beta t_{1}t_{2}-l\beta^{2}t_{2}^{2}+l{\dot{e}}_{2}(\tilde{d}-\dot{\theta })+k\alpha e_{2}t_{1}-k\beta e_{2}t_{2}\\ & =-l(\alpha t_{1}+\beta t_{2})2-k\beta e_{2}t_{2}+l{\dot{e}}_{2}(\tilde{d}-\dot{\theta })\\ & =-l(\alpha t_{1}+\beta t_{2})2-k\beta e_{2}\tanh(le_{2})+ke_{2}(\tilde{d}-\dot{\theta })+l{\dot{e}}_{2}(\tilde{d}-\dot{\theta })\\ & \leq -l(\alpha t_{1}+\beta t_{2})2-k\beta e_{2}\tanh(le_{2})+\left(k\left|e_{2}\right|+l\left|{\dot{e}}_{2}\right|\right)\left(\left|\tilde{d}\right|+\varepsilon_{1}\right)\\ & =-l(\alpha t_{1}+\beta t_{2})2-k\beta e_{2}\tanh(le_{2})+L \end{array}$$
where $L=\left(k\left|e_{2}\right|+l\left|{\dot{e}}_{2}\right|\right)\left(\left|\tilde{d}\right|+\varepsilon_{1}\right)$.Since $x\tanh(x)=x\left(e^{x}-e^{-x}\right)/\left(e^{x}+e^{-x}\right)\geq 0$, then $e_{2}\tanh(le_{2})\geq 0$, and only when $e_{1}=e_{2}=0$, ${\dot{V}}_{4}=0$ holds true. As the finite-time disturbance observer is stable and convergent; then, for any $\varepsilon_{2}>0$, there exists a finite variable $t_{3}$, so that $\left|\tilde{d}\right|<\varepsilon_{2}$ is valid. When $e_{1}\neq 0$ and $e_{2}\neq 0$, $-l{\left(\alpha t_{1}+\beta t_{2}\right)}^{2}-k\beta e_{2}\tanh(le_{2})<0$ is always true, and for any $\varepsilon_{3}>0$, there exists a finite variable $t_{4}$ such that ${\dot{V}}_{4}\leq 0$ is valid when $\left|e_{1}\right|\geq \varepsilon_{3}$. Therefore, the heading tracking error $e_{1}$ can reach a bounded set with a radius of $\varepsilon_{3}$ within a finite time and stay within it.According to the above analysis, every signal within the closed-loop system remains ultimately consistently bounded, and based on Lemma 1 and Lemma 2, the closed-loop system exhibits asymptotic stability, and its convergence speed depends on α,β,k,l. Therefore, when t→∞, $e_{1}\to 0$ and $e_{2}\to 0$. Thus, the proof of Theorem 2 is complete. □

**Remark** **6.***Since* $\tanh(x)\in \left[\begin{array}{cc} -1 & 1 \end{array}\right]$*, and according to* $\left|\tilde{d}\right|\leq \left|d\right|+\left|\hat{d}\right|$ *and (32), it is known that the adaptive course control law (35) is bounded and the amplitude is*(41)$$\begin{array}{ll} u_{c} & \leq \left|-\alpha \tanh(ke_{1}+le_{2})-\beta \tanh(le_{2})\right|+\left|-\hat{d}-f(x)-k_{3}\theta \right|\\ & \leq \alpha +\beta +\left|d\right|+\left|\tilde{d}\right|+k\left|\theta \right|+\left|f(x)\right| \end{array}$$

Therefore, from $u=u_{c}+\Delta u$ (9) and (28), it can be inferred that $\dot{\theta }$ is bounded. Therefore, Assumption 2 is reasonable.

## 4. Simulation Analysis

In this section, the adaptive control law $u_{c}$ is used for the heading tracking simulation of surface ships. It is compared and analyzed with the anti-disturbance control law $u_{a}$ and another robust one $u_{r}$ with an auxiliary dynamic system. The simulation object is chosen as the surface ship [15]. The model parameters are K=0.707,T=0.332,a=1. The input saturation amplitude of the rudder is $\delta_{\max}=35^{o}$.

Detailed information on the design methodology and parameters of the anti-disturbance control law $u_{a}$ may be found [15]. The robust control law $u_{r}$ with auxiliary dynamic system is designed as
(42)$$u_{r}=f(x)-\mu e_{1}-{\hat{d}}_{z}-k_{3}\theta$$
(43)$${\hat{d}}_{z}=z_{r}+Kx_{2}$$
(44)$${\dot{z}}_{r}=-K\left(u+f(x)\right)-K{\hat{d}}_{z}$$
where μ=0.4; K=1; $k_{3}=1$.

The desired heading is set to
(45)$$\psi_{d}=\left\{\begin{array}{c} 30^{o},0\leq t\leq 400\\ 20^{o},400<t\leq 800\\ -10^{o},t>800 \end{array}\right.$$

The design parameters are chosen as $\lambda_{0}=0.01$; $\lambda_{1}=0.8$; $\lambda_{3}=0.01$; $\delta_{0}=10^{-4}$; $k_{3}=1$; $L_{1}=0.01$; $L_{2}=10^{-5}$; α=3; β=0.125; k=0.2; l=2.5; and $\gamma =10^{-5}$.

The initial values for the simulation are set to ψ(0)=r(0)=0; $\sigma (0)=10^{-2}$; and $\theta (0)=10^{-3}$.

### 4.1. Case 1

The external disturbance [15] acting on the course control system is selected as
(46)$$\delta_{w}=0.02\sin(0.05t)+0.04\sin(0.08t)$$

The results from simulations are presented in Figure 4, Figure 5, Figure 6, Figure 7 and Figure 8. Figure 4 illustrates the heading tracking curves under different control laws, where $u_{c}$ developed in this paper can track the set heading quickly and accurately, while the robust control law $u_{r}$ results in larger fluctuations during the tracking process. Figure 5 shows the rudder angle response curve; the ship’s rudder angle under the designed control law is bounded and does not exceed the input amplitude (the blue dash line shown in Figure 5), which conforms to the actual steering situation. Figure 6 shows the ship’s bow angle response curve, where the ship’s bow angle under control laws $u_{c}$ and $u_{a}$ is smoother. Figure 7 shows the external disturbance and its estimation curve, indicating that the finite-time disturbance observer can accurately approximate the external disturbance. Figure 8 shows that the auxiliary dynamic system can effectively compensate for the rudder angle input deviation.

### 4.2. Case 2

To confirm the control law’s non-fragility against parameter perturbations in the system that is proposed in this paper, in this case, a ship mathematical model with model parameter perturbations is chosen for simulation, with the model parameters K=1.4×0.707, T=1.4×0.332, and a=1.4.

The external disturbance is chosen as
(47)$$\delta_{w}=0.04\sin(0.05t)+0.08\sin(0.08t)$$

The results from simulations are presented in Figure 9, Figure 10, Figure 11, Figure 12 and Figure 13, which indicate that the designed control law has similar control performance under parameter perturbation and large disturbance conditions as in Case 1, the ship’s rudder angle under the designed control law is also bounded and does not exceed the input amplitude (the blue dash line shown in Figure 10) indicating that the designed control law $u_{c}$ has strong non-fragility to system parameter perturbations. Although the control performance of the control algorithm in [15] has some fluctuations, it can still perform the tracking task well. However, the robust control law $u_{r}$ cannot complete the task any longer in this case. Figure 12 shows that the disturbance observer designed in this paper can still work normally under large disturbance conditions, which can ensure that the system has strong resistance to external disturbances.

## 5. Conclusions

This paper designs an innovative adaptive course control law for the ship course control system, which is influenced by external disturbances and input saturation. In this design, the external disturbance is appraised using a finite-time disturbance observer. The smooth saturation processing model and auxiliary dynamic system address the input saturation and rudder input deviation in the system. Furthermore, by integrating the hyperbolic tangent function control strategy with LaSalle’s Invariance Principle, an effective method for system stability analysis is designed. Lastly, the simulation results prove that the control law designed in this paper is effective and generally applicable.

## Figures and Tables

**Figure 1 sensors-24-04843-f001:**
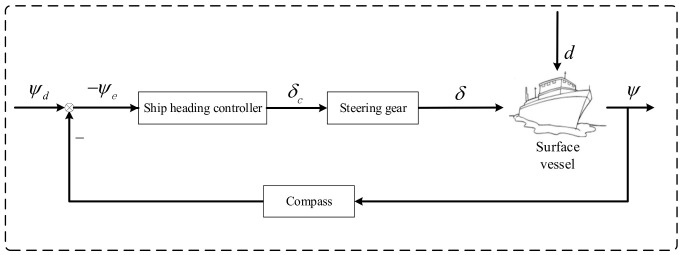
The ship course control system.

**Figure 2 sensors-24-04843-f002:**
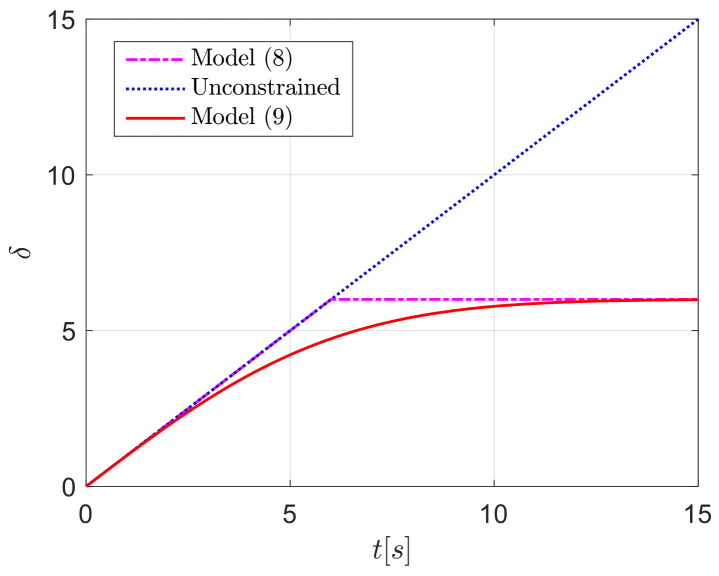
Comparison effect curves of different saturation models.

**Figure 3 sensors-24-04843-f003:**
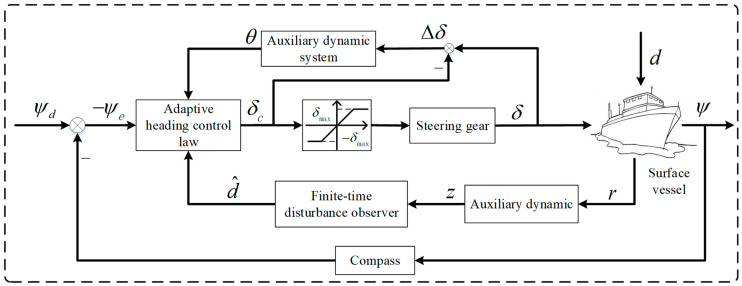
Adaptive course control system for surface ships.

**Figure 4 sensors-24-04843-f004:**
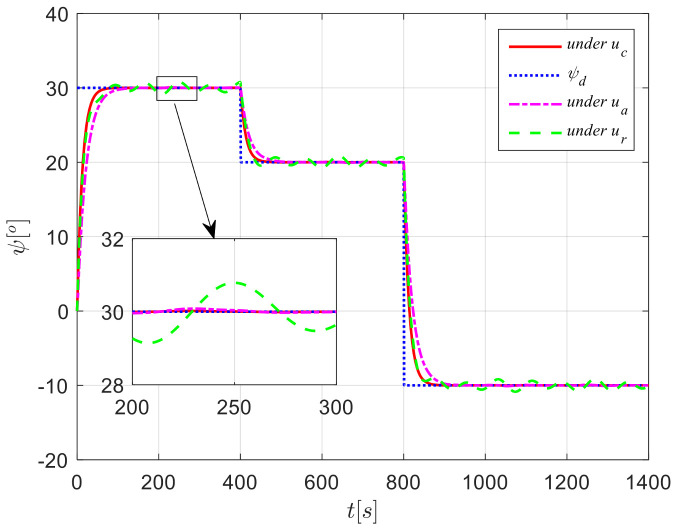
Course tracking curves in Case 1.

**Figure 5 sensors-24-04843-f005:**
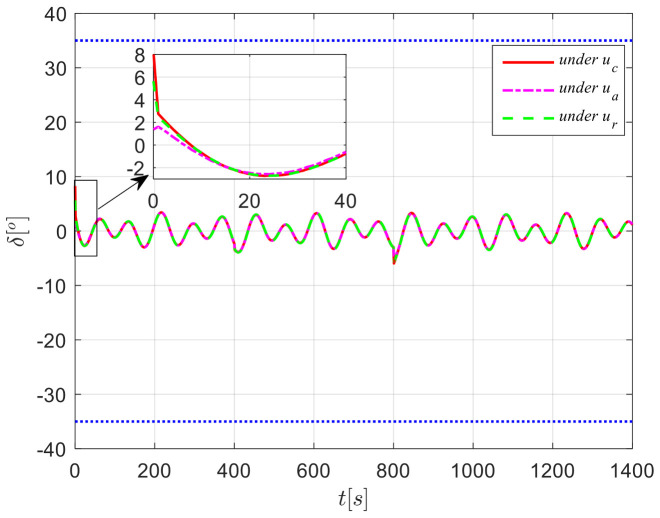
Rudder angle response curves in Case 1.

**Figure 6 sensors-24-04843-f006:**
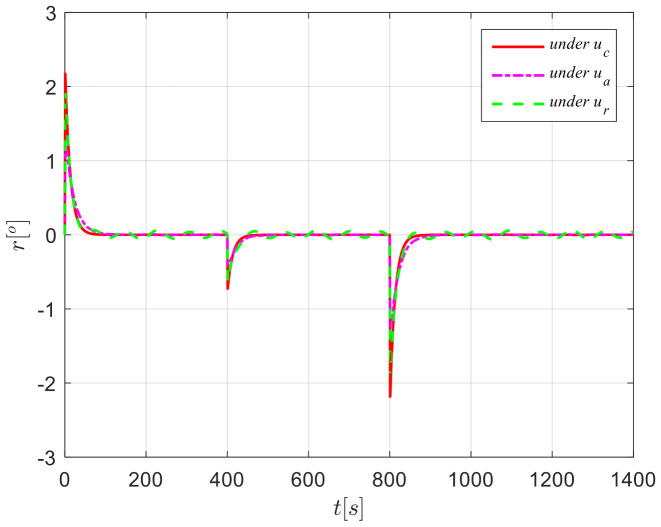
Heading response curves in Case 1.

**Figure 7 sensors-24-04843-f007:**
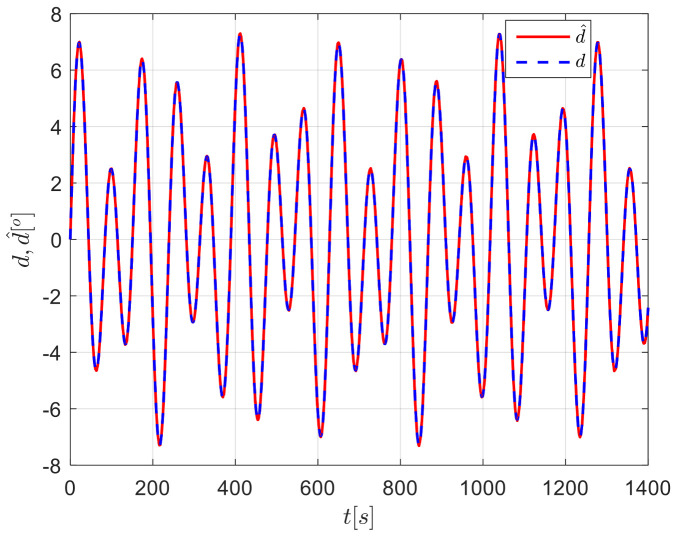
External disturbances and their approximation curves in Case 1.

**Figure 8 sensors-24-04843-f008:**
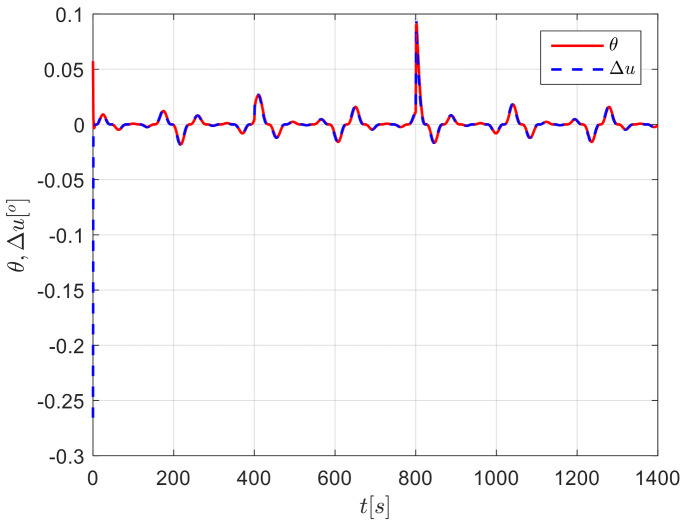
Input inconsistencies and their compensation curves in Case 1.

**Figure 9 sensors-24-04843-f009:**
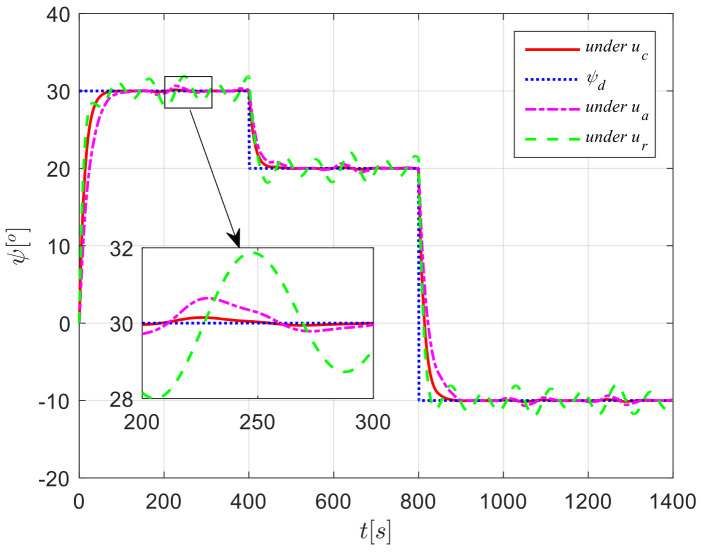
Course tracking curves in Case 2.

**Figure 10 sensors-24-04843-f010:**
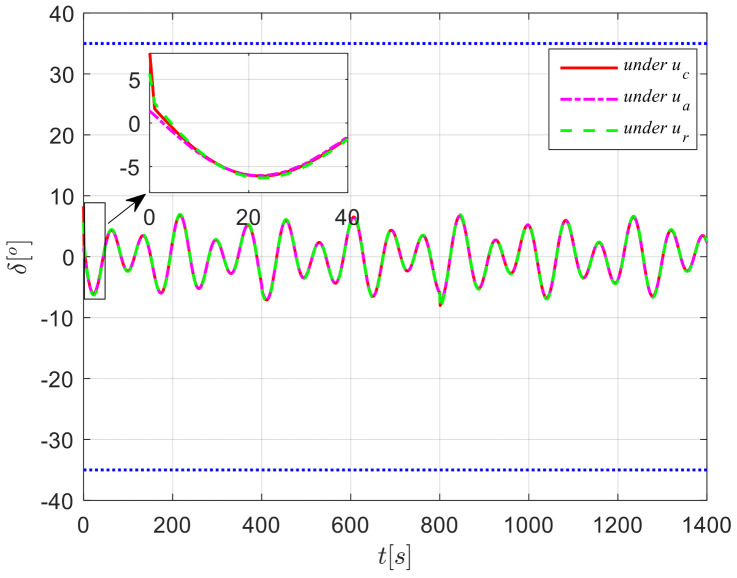
Rudder angle response curves in Case 2.

**Figure 11 sensors-24-04843-f011:**
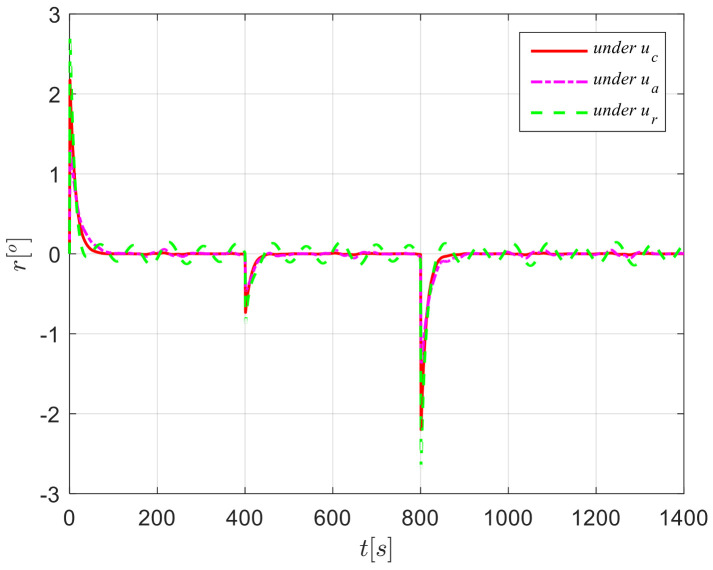
Heading response curves in Case 2.

**Figure 12 sensors-24-04843-f012:**
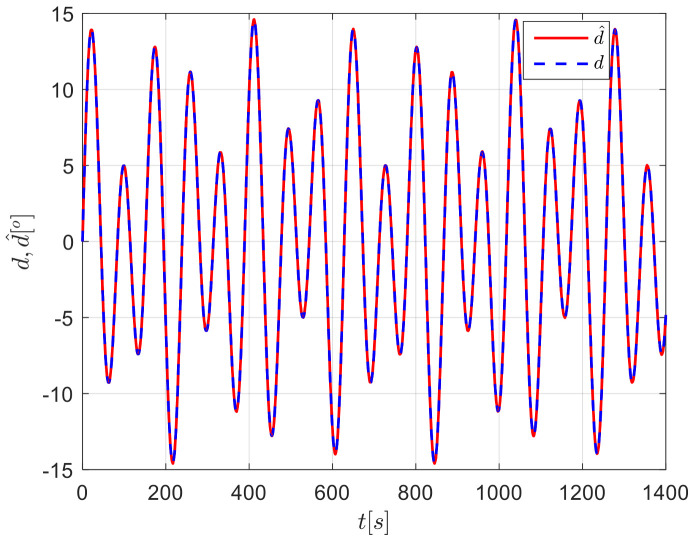
External disturbances and their approximation curves in Case 2.

**Figure 13 sensors-24-04843-f013:**
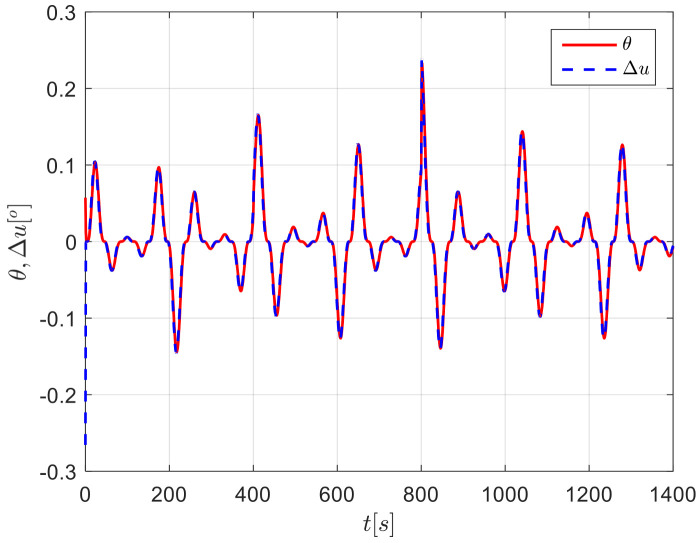
Input inconsistencies and their compensation curves in Case 2.

## Data Availability

No data were used for the descriptions in this paper.

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
