# Peer review of "Finite-Time Disturbance Observer-Based Adaptive Course Control for Surface Ships"

_sensors, 2024, doi:10.3390/s24154843_

Round 1

Reviewer 1 Report

Comments and Suggestions for Authors

This paper proposes an innovative finite-time disturbance observer based adaptive course control for surface ships. Through theoretical analysis and simulation verification, the effectiveness and stability of this strategy in addressing input saturation and external disturbances are demonstrated. The paper is highly innovative and systematic, though there is room for further optimization in some detailed descriptions and figure explanations. Overall, the paper makes a significant contribution to the field of ship control. I recommend minor revisions before acceptance.

1.Suggestion: Add a concise illustration to help readers intuitively understand the working principle of the ship course control system. For Assumptions 1 and 2, enhance their persuasiveness by citing more practical cases or experimental data.

2. Suggestion: Given the numerous symbols and variables used in the paper, consider providing a symbol table at the beginning or in an appendix for easy reference.

3. Suggestion: While the design and convergence proof of the finite-time disturbance observer are detailed, adding some practical application cases or simulation results would validate the method's effectiveness. For the convergence time of disturbance estimation errors, can you provide actual data or simulation results in addition to theoretical proof to illustrate its performance in practical applications?

4. Suggestion: The font size in Figure 2 is too small and increase the font size of the legend to be the same as the main text or one size smaller for better readability.

5. Suggestion: The legends obstruct parts of the data in some figures, such as Figure 11. Adjust the legends to avoid covering data points.

6. Suggestion: Ensure that the zoomed-in sections in Figure 3 have consistent coordinates with Figure 8 to better observe data changes.

7. Suggestion: The paper mainly verifies the control strategy's effectiveness through simulations. However, it lacks application examples or field test results in actual ship navigation environments. How does the proposed control strategy perform in practical sea conditions? How does it behave under different sea states and environmental conditions?

8. Suggestion: How are the control parameters proposed in the paper tuned? Is there a unified standard for parameter selection and tuning in different application scenarios?

Reviewer 2 Report

Comments and Suggestions for Authors

The article investigates the issue of ship course control under the influence of input saturation and external disturbances. This paper can be considered for potential publication after the following changes:

1. Further clarify the content and logical structure of the paper, clearly explaining the problems addressed, the purpose and significance of the research, and the value of the study.

2. The applicability of the research should be clearly defined. Due to differences in ship maneuverability, it should be specified whether the research applies to all ships.

3. In Section 3.2 of the article, to verify the robustness of the designed control law against system parameter perturbations, a ship mathematical model with parameter variations is selected for simulation. Please provide the reasons for this choice.

4. There are certain grammatical problems in the upper paragraphs of Remark 6.

5. Most of the derivations are correct, but in the derivation of some complex formulas (such as Formula 17 and Formula 26), there are minor issues with improper use of symbols or omissions in the calculation steps. It is recommended that the author double-check the derivation process of these formulas to ensure that each step is accurate.

Comments on the Quality of English Language

Minor editing of the English language is required.
